# Unilaterally Induced Quadriceps Fatigue during Sustained Submaximal Isometric Exercise Does Not Alter Contralateral Leg Extensor Performance

**DOI:** 10.3390/jfmk8020085

**Published:** 2023-06-19

**Authors:** Brian Benitez, Minyoung Kwak, Pasquale J. Succi, Joseph P. Weir, Haley C. Bergstrom

**Affiliations:** 1Department of Kinesiology and Health Promotion, University of Kentucky, Lexington, KY 40536, USA; bbe241@g.uky.edu (B.B.); mkw223@uky.edu (M.K.); pj.succi@uky.edu (P.J.S.); 2Department of Health, Sport & Exercise Sciences, University of Kansas, Lawrence, KS 66045, USA; joseph.weir@ku.edu

**Keywords:** non-localized muscle fatigue, cross-over effects, performance fatigability

## Abstract

This study investigated the effects of fatiguing unilateral exercise on the ipsilateral, exercised, and contralateral, non-exercised limb’s post-exercise performance in males and females. Ten males and ten females performed a fatiguing, unilateral isometric leg extension at 50% maximal voluntary isometric contraction (MVIC) force. Prior to and immediately after the fatiguing tasks, MVICs were performed for the exercised and non-exercised limb, with surface electromyographic (sEMG) and mechanomyography (sMMG) amplitude (AMP) and mean power frequency (MPF) recorded from each limb’s vastus lateralis. There were no fatigue-induced, sex-dependent, differences in time to task failure (*p* = 0.265) or ipsilateral performance fatigability (*p* = 0.437). However, there was a limb by time interaction (*p* < 0.001) which indicated decreases in MVIC force of the ipsilateral, exercised (*p* < 0.001), but not the contralateral, non-exercised limb (*p* = 0.962). There were no sex-dependent, fatigue-induced differences in neurophysiological outcomes between the limbs (*p* > 0.05), but there was a fatigue-induced difference in sEMG MPF (*p* = 0.005). To summarize, there were no differences in fatigability between males and females. Moreover, there was insufficient evidence to support the presence of a general crossover effect following submaximal unilateral isometric exercise. However, independent of sex, the neurophysiological outcomes suggested that competing inputs from the nervous system may influence the performance of both limbs following unilateral fatigue.

## 1. Introduction

Exercise-induced fatigue can be characterized by a transient reduction in the force-generating capacity of a muscle, typically resulting from sustained or repeated contractions [1]. Although force capacity is ultimately limited by the output of the active muscle(s), exercise-induced fatigue is a multifactorial phenomenon that can arise from both central and peripheral factors [1]. Central factors commonly include mechanisms proximal to the neuromuscular junction, wherein fatigability depends on the competing influences between excitatory and inhibitory inputs from the nervous system (i.e., group III/IV afferent feedback from the muscles to the nervous system) [1,2,3]. Conversely, peripheral factors occur distal to the neuromuscular junction and primarily concern the contractile function of the involved muscle (i.e., cross-bridge kinetics, muscle metabolism, etc.) [1,4,5]. Indeed, both central and peripheral factors share a common point of overlap near the neuromuscular junction where alterations in the intramuscular environment of the working muscle may elicit type III/IV afferents signaling to reduce the neural drive to the muscle [1,6]. Thus, identifying the primary mechanisms for the transient reduction in the force generating capacity of a muscle can be challenging. 

The cross-over effect refers to a phenomenon in which unilateral, exercise-induced fatigue in one muscle group transiently impairs the neuromuscular function of the non-exercised, contralateral, homologous muscle group [7]. The cross-over effect is particularly intriguing because it offers unique insights into mechanisms of fatigue, occurring proximal to the neuromuscular junction (i.e., central fatigue). However, the systemic effect of unilateral fatigue on the force production of the non-exercised contralateral, homologous muscle groups demonstrates varying responses ranging from no change [8,9,10,11,12,13] to decreases [14,15,16,17,18], and even facilitation [19,20] in the force generating capacity of the non-exercised contralateral, homologous muscle groups. A recent meta-analysis by Behm et al. [21] found no clear evidence of a general crossover effect; however, the impact of sex was difficult to ascertain as only three studies identified directly compared sex-related crossover effects. Martin and Rattey [16] and Ye et al. [18] reported greater contralateral fatigue for males compared to females following sustained, maximal, isometric leg extensions. Under similar conditions, however, Doix et al. [8] found greater vastus lateralis surface electromyographic (sEMG) activity in females compared to males, despite no significant crossover effect for either sex. It is well established that the nature and magnitude of exercise-induced fatigue differs between sexes [22,23]. That is, females are typically more fatigue-resistant than males during sustained, submaximal, isometric contractions, reflected by greater times-to-task failure (TTF) [22]. Yet, despite reported differences between males and females during the examination of fatiguing tasks, the effect of fatigue on the non-exercised, contralateral homologous muscle groups has not been widely examined in males and females outside of maximal exercise conditions [21]. Thus, the extent and significance of the cross-over effect in males and females remain unclear.

At present, little is known about the mechanisms that contribute to the cross-over effect. It has been theorized that metabolic perturbations within the working muscle may elicit a response from group III/IV afferent fibers, reducing neural drive to the contralateral limb in the absence of peripheral fatigue [24,25]. Martin and Rattey [16] reported greater centrally-mediated contralateral fatigue for males compared to females, indicated by greater decreases in both MVIC force and voluntary activation (i.e., 1-superimposed twitch/resting twitch × 100) of males. These findings suggest that there may be sex-dependent differences in the way the nervous system responds to unilateral fatigue. Neuromuscular responses derived from sEMG signals may aid in elucidating the neurophysiological underpinnings of the cross-over effect. Indeed, contained within the sEMG signal is a mixture of information regarding the synaptic inputs received by the motor neurons and the muscle fiber’s electrical characteristics [26]. It is generally accepted that the amplitude (AMP) characteristics of the sEMG signal reflect a composite measure of muscle excitation, containing information regarding motor unit recruitment, firing rates, and synchronization [26], whereas the mean power frequency (MPF) characteristics of the sEMG signal reflect the action potential conduction velocity along the sarcolemma [27]. However, without sophisticated decomposition procedures employed under specific exercise protocols (i.e., trapezoidal isometric contractions), it is not possible to delineate between motor unit recruitment strategies using solely the AMP characteristics of the sEMG signal [26]. Alternatively, features derived from surface mechanomyographic (sMMG) signals, which has been described as the mechanical counterpart of the sEMG signal [28], have been utilized to partition fatigue-induced motor unit recruitment strategies into recruitment and rate coding. Indeed, the AMP characteristics of the sMMG signal has been suggested to contain information regarding motor unit recruitment, while the frequency characteristics are believed to reflect the global firing rate of unfused, activated, motor units [29]. Thus, the simultaneous examination of sEMG and sMMG features may offer a unique perspective into the motor unit recruitment strategies underlying the cross-over effect.

The primary purpose of the present study was to investigate the effect of submaximal, fatiguing, isometric leg extensions on MVIC force of the ipsilateral, exercised, and contralateral, non-exercised, limb in males and females. Secondarily, sEMG and sMMG AMP and MPF measures recorded during the MVIC trials were used to inform potential neurophysiological mechanisms related to a cross-over effect. It was hypothesized that the females would demonstrate a longer time to task failure than the males, and there would be greater fatigue-induced decreases in the MVIC force of the exercised limb for the males compared to females, but no change in the MVIC force for the non-exercised limb regardless of sex. It was also hypothesized that there would be sex-dependent fatigue-induced differences between limbs for each respective neurophysiological outcome (sEMG AMP, sEMG MPF, sMMG AMP, and sMMG MPF). 

## 2. Materials and Methods

### 2.1. Experimental Approach

The effect of fatiguing, unilateral, isometric leg extensions on changes in MVIC force and sEMG and sMMG AMP and MPF of the ipsilateral, dominant, exercised, and contralateral, non-dominant, non-exercised limb were examined in healthy, recreationally trained males and females. Testing procedures involved performing isometric muscle actions on a modified leg extension machine (Body-Solid, GLCE365, Forest Park, IL, USA) instrumented with a load cell (Honeywell Model 41 Precision Low Profile Load Cell, Charlotte, NC, USA). Furthermore, neuromuscular responses (AMP and MPF) derived from sEMG and sMMG signals were recorded from the vastus lateralis of the exercised and non-exercised limb to inform underlying neurophysiological mechanisms related to the cross-over effect.

This study was approved by the University of Kentucky Institutional Review Board for Human Subjects (IRB #73699, approved 18 October 2021) meeting the ethical standards of the Helsinki declaration. Importantly, data in the present investigation were part of a larger study that included multiple dependent and independent variables. The data in the current investigation have not been previously published. 

### 2.2. Participants

A convenience sample of 23 recreationally active males and females with previous resistance training experience were initially recruited. However, 3 participants dropped out of the study due to personal reasons unrelated to the study and thus, the final sample size included 20 participants. Specific participant characteristics can be found in Table 1. Participants were required to have resistance training experience of at least 1 year and to have abstained from any strenuous physical activity for 48 h prior to testing. Additional exclusion criteria were based upon illness or any contraindications to physical activity identified using a health history questionnaire. All participants were informed of the risks and benefits of the study, completed a health history questionnaire, and signed a written informed consent document before participating in this study. 

### 2.3. Familiarization Session

Despite all of the participants reporting familiarity with performing exercise on a leg extension machine, it is uncommon for individuals to regularly perform fatiguing, isometric leg extension in their own training. Thus, to reduce the potential deleterious effects of participant unfamiliarity with the protocol, a familiarization session was included. The procedures for the familiarization session included submaximal and maximal, bilateral and unilateral isometric leg extensions. Additionally, participants performed non-fatiguing, submaximal, isometric holds to a target force with a visual aid in the form of a horizontal on-screen force tracing. Participants were instructed to match the on-screen force with a target force for 10–20 s to ensure that they could perform the experimental task.

### 2.4. Experimental Session

The experimental sessions started with a standardized warm-up consisting of three separate, submaximal unilateral dominant, unilateral non-dominant, and bilateral isometric contractions (approximately 30%, 50%, and 80% of MVIC) of the leg extensors. Participants then performed pre-testing, which included 3–4 separate 3 s maximal bilateral, dominant-unilateral, and non-dominant-unilateral isometric leg extensions at a joint angle of approximately 120° (180° corresponding to full extension) to determine peak force. The testing order was randomized, and participants were given 2–4 min of rest between trials. The trial which resulted in the greatest peak force was used as the pre-test MVIC force. After pre-testing, participants were given 5 min of rest before performing a fatiguing, submaximal, dominant-unilateral isometric leg extension at a force that corresponded with 50% of their highest dominant-unilateral pre-test MVIC. During the fatiguing task, participants were provided with a visual aid in the form of a horizontal on-screen force tracing. Task failure was defined as an inability to maintain the target force tracing (i.e., within 5% of the target force), despite strong verbal encouragement. Immediately following the fatiguing task, participants completed post-testing, which included a total of three 3 s, maximal bilateral, dominant-unilateral (ipsilateral, exercised limb), and non-dominant-unilateral (contralateral, non-exercised limb) isometric leg extensions, in a randomized order. Participants received no rest between post-test trials and all trials were collected within 15 s of task failure.

During the experimental trials, sEMG signals were recorded using a bipolar electrode arrangement (ST-50 AccuSensor 38 mm diameter, silver/silver chloride, Lynn Medical, Wixom, MI, USA), placed on the vastus lateralis of the exercised and non-exercised limb, with an inter-electrode distance of 30 mm. The sMMG signals were detected with an accelerometer (Entran EGAS FT 10, bandwidth 0–200 Hz, dimensions: 1.0 × 1.0 × 0.5 cm, mass 1.0 g sensitivity 10 mV g^−1^) placed between the bipolar sEMG electrode arrangement, using double-sided adhesive tape. Electrode placement was made according to recommendations from the Surface Electromyography for the Non-invasive Assessment of Muscles (SENIAM) project (http://www.seniam.org/, accessed on 1 November 2021). The sEMG and sMMG signals were sampled at 1 kHz using a 16-bit analog analog-to-digital converter (Model MP150, BIOPAC Systems, Inc., Santa Barbara, CA, USA).

### 2.5. Signal Processing

The sEMG and sMMG signals were processed using an in-lab designed MATLAB program. During the pre-test and post-test MVICs, a 1 s epoch corresponding with the middle 1/3 of the contraction was isolated for signal processing. The sEMG signals were differentially amplified (EMG 100 c, BIOPAC Systems, Inc., Santa Barbara, CA, USA, bandwidth = 10–500 Hz, gain: ×2000) and digitally bandpass filtered (zero-phase shift fourth-order Butterworth) at 10–499 Hz. The sMMG signals were amplified with an in-line amplifier (gain: 200) and digitally bandpass filtered (zero-phase shift fourth-order Butterworth) at 5–100 Hz. Following bandpass filtering, sEMG and sMMG signals were rectified, and a root mean square envelope was generated (50 millisecond moving window) to identify the AMP characteristics of the signals. A fast Fourier transform (FFT—Hamming window processing) was applied to the filtered signals to identify the frequency characteristics of the power spectrum periodogram. 

All statistical analyses were conducted using the un-normalized sEMG and sMMG features, with the pre-test value included in the model as a covariate. The inclusion of a pre-test covariate results in adjusted estimates that account for natural variability in the signal while preserving the naturally occurring variance, occupying a similar role to normalization [30]. However, to facilitate the comparison of our data to other literature, sEMG and sMMG features normalized to pre-test MVIC are also presented descriptively in Table 2.

### 2.6. Statistical Analysis

For the present study, sample size is justified based on feasibility [31], and thus, no formal power analysis was performed. As many participants as possible were recruited given the constraints on the investigators’ time and resources. Thus, efforts have been undertaken to ensure that data are as easy as possible to meta-analytically aggregate in the future. All statistical analyses were performed using ‘R’ software (v 4.0.2; R Core Team, https://www.r-project.4org/, accessed on 1 November 2021).

To investigate the reliability of the measurements, separate reliability analyses were conducted for the pooled pre-test values of the dominant- and non-dominant limb’s pre-test MVIC force, sEMG AMP, sEMG MPF, sMMG AMP, and sMMG MPF, respectively. Specifically, the three highest MVIC pre-test values of the dominant and non-dominant trials were used for the reliability analysis, which included a repeated measures analysis of variance (ANOVA) to assess systematic error, as well as calculation of intraclass correlation coefficients (ICCs), 95% confidence intervals (CI_95%_), standard error of the measurement (SEM), coefficient of variation (CV), and minimal detectable change (MDC) using a two-way random effects model (ICC2k) [32]. Reliability was assessed using the “simplyAgree” package.

An independent sample t-test was first utilized to compare TTF for the fatiguing tasks between males and females. To investigate the presence of a cross-over effect, the MVIC forces from both limbs of males and females prior to and immediately following a fatiguing isometric contraction of the exercised limb were then compared utilizing linear mixed-effect (LME) models fit with restricted maximum likelihood estimation (REML). Fixed effects, and interactions thereof, were included for sex, limb, and time. Random intercepts were included per participant to account for repeated measures. To inform underlying neurophysiological mechanisms related to any changes in MVIC force, or lack thereof, separate, LME models fit with REML were constructed for all neurophysiological outcomes (sEMG and sMMG AMP and MPF). The post-test values for each outcome variable were included in each respective model as the dependent variable, with fixed effects and interactions thereof included for sex and limb. Pre-test values were included as a covariate, with random intercepts included per participant to account for repeated measures. Although neither pre-test nor within-group inferential statistics were calculated, we descriptively presented within-group changes to help contextualize our findings. 

To address our primary research questions, null-hypothesis significance testing (α = 0.05) was performed to evaluate the presence of main or interaction effects using the “lmerTest’’ package. Specifically, ANOVA tables were generated using Type III sum of squares and Satterthwaite’s method of estimating denominator degrees-of-freedom and F-statistics [33]. Pending any significant interactions or main effects, contrasts were produced using CI_95%_ to support inferences regarding statistical differences. To further facilitate practical interpretation of our results, unstandardized effect sizes are reported, wherever possible, which is in line with general recommendations for reporting effect sizes [34]. Prior to performing any tests or extracting model estimates, the quality of model fit was assessed using the “performance” package (see https://osf.io/cmzqx, accessed on 2 May 2023). Finally, to reconcile concerns regarding the influence of suspected outliers in the data, a leave-one-out sensitivity analysis was performed for all models concerning neurophysiological outcomes (see https://osf.io/cmzqx, accessed on 2 May 2023).

## 3. Results

### 3.1. Reliability Outcomes

The results of the reliability analysis indicated excellent reliability (ICC > 0.90) for all variables tested [35]. The specific outputs for each analysis with ICC, SEM, and CV are presented in Table 3. 

### 3.2. Time-to-Exhaustion and Maximal Voluntary Force Outcomes

For TTF, an independent samples *t*-test indicated that there were no significant differences between males and females (mean difference (MD) = −11.94 s [CI_95%_: −33.74, 9.87]; t = 1.15; *p* = 0.265) (Figure 1A,B). 

For MVIC force, the LME model indicated no significant three-way interaction between Sex, Limb, and Time (F(54,1) = 0.612; *p* = 0.437), which indicated that the change in MVIC force between limbs was not mediated by sex (MD = −8.06 kgf [CI_95%_: −23.72, 7.57]) (Figure 2A,B). There was, however, a significant two-way interaction between Limb and Time (F(54,1) = 61.416; *p* < 0.001), which indicated a significant difference for the change in MVIC force between limbs, adjusted for sex (MD = −39.69 kgf [CI_95%_: −39.69, −24.05]) (Figure 2C,D). Follow-up contrasts also indicated changes in MVIC force for the dominant limb (Post-Test—Pre-Test: MD = −33.21 kgf [CI_95%_: −40.46, −25.96]; t = −12.151; *p* < 0.001), but not the non-dominant limb (Post-Test—Pre-Test: MD = −1.34 kgf [CI_95%_: −8.72, 6.04]; t = −0.483; *p* = 0.962). Finally, there was a significant main effect of Sex (F(18,1) = 10.503; *p* = 0.005), indicating greater MVIC force values for the males compared to females (Males—Females: MD = 27.88 kgf [CI_95%_: 10.14, 45.63]) (Figure 2E,F).

### 3.3. Neurophysiological Outcomes

The leave-one-out sensitivity analysis did not meaningfully change the interpretation of our primary research question for any models concerning neurophysiological outcomes (see https://osf.io/cmzqx, accessed on 2 May 2023). Thus, inferences and parameter estimations were based upon the model iteration containing all participant data. 

For sEMG AMP, the LME model indicated that the covariate, pre-test sEMG AMP, was significantly related to the post-test sEMG AMP (F(29.674,1) = 40.351; *p* < 0.001; r = 0.76). However, there was no significant two-way interaction between Sex and Limb (F(17.904,1) = 0.068; *p* = 0.797), indicating that the difference in fatigue between limbs for sEMG AMP did not depend on sex (MD = 19.86 uV [CI_95%_: −139.76, 179.48]) (Figure 3A,B). No notable main effects for Limb (F(19.408,1) = 0.040; *p* = 0.844) or Sex were identified (F(16.937,1) = 0.009; *p* = 0.927). 

For sEMG MPF, the LME model indicated that the covariate pre-test sEMG MPF was significantly related to post-test sEMG MPF (F(28.222,1) = 24.058; *p* < 0.001; r = 0.54). However, there was no significant two-way interaction between Sex and Limb (F(18.939,1) = 0.993; *p* = 0.332), indicating that differences for the effect of fatigue between limbs for sEMG MPF did not depend on sex (MD = 6.13 Hz [CI_95%_: −6.75, 19.00]) (Figure 4A,B). There was, however, a notable significant main effect for Limb (F(17.985,1) = 26.696; *p* < 0.001), which suggests a greater effect of fatigue on sEMG MPF of the exercised limb compared to the non-exercised limb, when adjusted for sex (MD = −21.95 Hz [CI_95%_: −21.95, −9.26] [CI_90%_: −20.84, −10.37]) Figure 4C,D). No notable main effect for Sex was identified (F(17.446,1) = 1.286; *p* < 0.272). 

For sMMG AMP, the LME model indicated that the covariate, pre-test sMMG AMP, was significantly related to post-test sMMG AMP (F(29.336,1) = 24.058; *p* < 0.001; r = 0.81). However, there was no significant two-way interaction between Sex and Limb (F(18.341,1) = 0.010; *p* = 0.923), indicating that the difference in fatigue between limbs for sMMG AMP did not depend on sex (MD = 0.01 m·s^−2^ [CI_95%_: −0.17, 0.18] [CI_90%_: −0.14, 0.15]) (Figure 5A,B). No notable main effects for Limb (F(19.824,1) = 0.587; *p* = 0.453) or sex (F(23.093,1) = 0.406; *p* = 0.530) were identified. 

For sMMG MPF, the LME indicated that the covariate pre-test sMMF MPF was significantly related to the post-test sMMG MPF (F(35,1) = 61.303; *p* < 0.001; r = 0.77). However, there was no significant two-way interaction between Sex and Limb (F(35,1) = 0.008; *p* = 0.930), indicating that the difference in fatigue between limbs for sMMG MPF did not depend on sex (MD = 0.33 Hz [CI_95%_: −7.24, 7.90] [CI_90%_: −5.97, 6.63]) (Figure 6A,B). No notable main effects for Limb (F(35,1) = 4.076; *p* = 0.051) or Sex (F(35,1) = 0.389; *p* = 0.537) were identified. 

## 4. Discussion

This study investigated the effects of fatiguing submaximal unilateral exercise on the ipsilateral and contralateral limb’s post-exercise performance across sex. Contrary to our hypothesis, females did not demonstrate longer TTF than males. Further, despite observing fatigue-induced changes in the MVIC force of the dominant exercised limb but not the non-dominant, non-exercised limb, we failed to demonstrate convincing evidence to suggest that the magnitude of fatigue-induced changes depended on sex. Results of the secondary analysis for neurophysiological outcomes were also contrary to our hypotheses. Specifically, there were no fatigue-induced, sex-dependent differences between limbs for any neurophysiological outcome (sEMG AMP, sEMG MPF, sMMG AMP, and sMMG MPF). There was, however, a notable main effect of limb for sEMG MPF, which suggested that fatigue differentially affected the exercised and non-exercised limbs. 

### 4.1. Time-to-Task Failure and Maximal Voluntary Isometric Force Outcomes

In the present study, there was no evidence to indicate that the females were more fatigue-resistant than the males. Males and females differ in both anatomy and physiology, usually resulting in appreciable sex differences in neuromuscular performance and fatigability [22,23]. Differences in muscle perfusion between males and females is thought to be a primary contributor to sex differences in muscle fatigue and performance [22,23]. That is, females generally exhibit greater muscle perfusion than males for some muscle groups due to differences in vasoconstriction and greater capillarization of the muscle bed [36,37,38]. In the present study, however, there were no differences in TTF (MD = −11.94 s [CI_95%_: −33.74, 9.87]) or performance fatigability between males and females. There is some evidence to suggest that sex differences in fatigability become smaller or cease to exist for fatiguing isometric exercise performed at intensities moderate to high relative intensities (≥50%) [39]. Moreover, intramuscular occlusion of blood flow has been reported to occur at lower relative intensities (<50% MVIC) [40,41], regardless of sex, which would theoretically negate much of the oxidative advantage that females have been shown to have over males in the current study. Thus, it is possible that during the fatiguing trials, force remained sufficiently high such that oxygen delivery was limited throughout the protocol similarly for males and females, resulting in no differences in TTF or performance fatigability of the exercise limb. 

The effect of sex on fatigue-induced alterations of the non-exercised, contralateral limbs has not been widely examined in literature outside of a few studies [21], which demonstrated mixed results. Martin and Rattey [16] and Ye et al. [18] both reported greater contralateral fatigue for males compared to females. Conversely, Doix et al. [8] reported that males experienced greater fatigue in the exercised limb but reported no significant cross-over effects for either sex. Most recently, Voskuil et al. [20] reported increased contralateral MVIC force of the wrist flexors for the males but no change for the females following a hold to task failure at 50% of MVIC. In the present study, there was no evidence to suggest that fatigability of the exercised limb was sex-dependent (%Δ = Males: −36.87% vs. Females: −41.01%), nor were there significant fatigue-induced alterations in MVIC force in the non-exercised contralateral limb, regardless of sex (%Δ = −2.72%). Our findings are in line with the meta-analytical findings from Behm and colleagues [21], which found trivial evidence of cross-over force deficits associated with unilateral fatigue with no impact of sex [β = −0.02 (CI95% = −0.14, 0.09)]. Though the intensity of fatiguing protocol has not been found to moderate the presence of a cross-over effect [β = ~0.00 (CI95% = −0.0001, ~0.00)] [21], the relationship between sex-dependent fatigability and intensity of contraction has not been fully elucidated, as only 13 studies identified for the meta-analysis reported the inclusion of females, and only three studies [8,16,18] directly compared males and females. For example, both Martin and Rattey [16] and Ye et al. [18] utilized sustained [16] and intermittent [18] maximal isometric contractions for their fatiguing conditions and indicated sex-dependent differences in magnitude of contralateral fatigue. Conversely, the present study utilized fatiguing contractions maintained at a submaximal intensity (50% MVIC) and found no evidence of cross-over force deficits. Thus, it is possible that sex-dependent cross-over effects may also depend upon the intensity with which the task is being performed. However, it is also possible that the disagreement between studies may be explained by variability in perceptual responses and effort associated with the varying methodological approaches to the question. 

It has been previously suggested that the performance decrements during fatiguing tasks may be attributable to mental fatigue, which may alter perceptions of effort during subsequent performance tests [42]. Marcora [43] proposes that the perception of effort is the “…conscious sensation of how hard, heavy, and strenuous a physical task is”, conceptualizing perception of effort as one’s appreciation of task difficulty. Previous literature has shown that mentally fatiguing tasks alone can impede physical performance by increasing perceptions of effort, resulting in reduced performance outcomes [42]. Moreover, enduring an isometric contraction to the point of task failure is uncomfortable, sometimes painful, and requires focus and concentration to sustain task demands. Thus, it is possible that mental fatigue associated with certain physical activities may influence perceptions of fatigue, explaining some of the variability in the literature around the cross-over effect. Of course, these cognitive impairments are inherently underpinned by physiology. Gandevia et al. [43] provided evidence for this using an ischemic block to unilaterally deafferent a hand and reported greater discrepancies in perceived movement of the contralateral hand with the intensity level of the motor command. Though it is unlikely that any single physiological mechanism offers a clear explanation for cross-over fatigue, the global sensory tolerance limit suggests exercise ceases or is reduced based on the sum of all neural feedback and feedforward signals [43]. This global negative feedback loop is enhanced by sensory effects from muscles directly (i.e., exercised muscle group) or indirectly (i.e., respiratory) involved in exercise [44] Thus, the ability to tolerate subsequent performance tasks following a fatiguing protocol may explain variability within the literature about the cross-over effect. In this study, it is possible that the magnitude of metabolic perturbations and afferent feedback for the hold to task failure at 50% MVIC were not sufficient to alter performance of the non-exercised limb.

### 4.2. Neurophysiological Outcomes

It is well recognized that the sEMG power spectrum shifts towards lower frequencies during fatiguing isometric contractions [27]. This shift has been previously attributed to a decrease in the muscle fibers’ conduction velocity and/or synchronous firing of motor units [27]. Moreover, parameters derived from the frequency spectrum have been used as indicators of local muscle fatigue during isometric contractions [27]. Though we hypothesized sex-dependent changes for all neurophysiological outcomes, the absence of any sex differences for TTF suggested that fatigue affected the sEMG MPF of both males and females similarly. There were, however, meaningful differences for the effects of fatigue between limbs, which was attributed to greater downward shifts in MPF of the exercised limb (%Δ = −18.76%) compared to the non-exercised limb (%Δ = −3.12%). In a similarly designed study, Kawamoto et al. [15] reported contralateral deficits in the non-exercised leg with no significant change in sEMG median frequency of the vastus lateralis, which suggested that alterations reported in the non-exercised limb’s motor performance may have been mediated by factors unrelated to motor unit conduction velocity and/or synchronicity. 

In the present study, it was found that fatigue affected the sEMG AMP of the exercised and non-exercised limbs similarly between males and females (MD = 19.86 uV [CI_95%_: −139.76, 179.48] [CI_90%_: −111.87, 151.56]). That is, sex did not meaningfully influence the magnitude of change for sEMG AMP. Moreover, there were no effects for limb to suggest that fatigue differentially affected sEMG AMP of both limbs. These findings are contrary to the findings of Kawamoto et al. [15] who found evidence for contralateral deficits in the non-exercised leg, but only significant reductions in the sEMG AMP of the exercised vastus lateralis. One potential explanation for the lack of agreement between our findings and those of Kawamoto et al. [15] may be related to the inherent variable nature of the sEMG AMP characteristics. The sEMG signal is highly variable, with many physiological [43] factors that can influence the magnitude (AMP) of the signal. In fact, sEMG AMP can be summarized as the sum of all excitatory and inhibitory inputs from the motor neuron [26]—containing a combination of information related to motor unit recruitment, rate coding, and synchronization [25]. It is possible that a portion of the signal’s variable nature may be attributed to the complex relationship between modulation of motor unit recruitment, rate coding, and/or synchronization necessary to sustain task demands. That is, similar levels of sEMG AMP do not necessarily indicate that motor unit behavior (recruitment, discharge rates, and/or synchronization) are equivalent between muscles [45].

The simultaneous examination of sEMG AMP alongside sMMG AMP and MPF has been previously utilized during fatiguing exercise to partition fatigue-induced motor unit recruitment strategies into recruitment and rate coding [29]. Specifically, under certain conditions, the sMMG AMP may reflect global indices of motor unit recruitment, whereas sMMG MPF may reflect the global discharge rates of activated, unfused motor units [29]. In the present study, it was found that fatigue affected the sMMG AMP (MD = 0.01 m·s^−2^ [CI_95%_: −0.17, 0.18] [CI_90%_: −0.14, 0.15]) and MPF (MD = 0.330 Hz [CI_95%_: −7.24, 7.90] [CI_90%_: −5.970, 6.629]) of the exercised and non-exercised limbs similarly across sex. That is, sex did not meaningfully influence the magnitude of change in motor unit recruitment or rate coding. Moreover, there were no effects for limb to suggest that fatigue differentially affected sMMG AMP and MPF of both limbs. Previous research has suggested [46,47] that there may be inherent properties (soma size, membrane permeability, capacitance) about the motor unit which dictate neuronal recruitment thresholds and the motor unit’s behavior after threshold has been reached. Thus, even if sEMG AMP values are seemingly equivalent, alterations to the intramuscular environment could alter factors related to recruitment, rate coding, and/or synchronization. For example, during sustained, submaximal, fatiguing contractions, neural drive to a muscle must increase, leading to synaptic input that is common to more than one neuron [45]. With the onset of localized muscle fatigue, the changes in neural drive may alter synchronization of motor units discharge patterns [48], and this might be explained by the commonality in the pre-synaptic input to motor units. Moreover, it has been theorized that metabolic perturbations within the working muscles may elicit a response from III/IV afferent fibers, altering neural drive to the contralateral, non-exercised, limb in the absence of peripheral fatigue [19,24,25]. It is possible that the absence of a cross-over effect despite no difference in responses to unilateral fatigue for sEMG AMP between limbs in the present study may be explained by differences in synchronization due to peripheral fatigue of the exercised limb, as reflected by the downward shifts in sEMG MPF for only the exercised limb. 

### 4.3. Limitations

This study examined only recreationally trained college-aged participants; therefore, it is uncertain if the present results can be extrapolated to clinical populations, untrained subjects, and individuals of different ages. Finally, data in the present investigation were part of a larger study and thus, pre-testing and post-testing MVIC trials included a bilateral MVIC in addition to the dominant and non-dominant MVICs. It is thus possible that a portion of variance could be attributed to instances where the bilateral MVIC was performed prior to the non-dominant limbs MVIC. 

## 5. Conclusions

In summary, these data suggested that, following a fatiguing unilateral intervention, the contralateral homologous muscle groups motor performance was unaffected. However, the neurophysiological outcomes suggested that the performance of the ipsilateral exercised and contralateral non-exercised limb after unilateral fatigue may be mediated, at least in part, by competing influences between excitatory and inhibitory inputs from the nervous system. Future studies should include more direct comparisons between males and females under various intensities (maximal vs. submaximal) and types (intermittent vs. continuous) of contractions to further explore the cross-over effect. 

## Figures and Tables

**Figure 1 jfmk-08-00085-f001:**
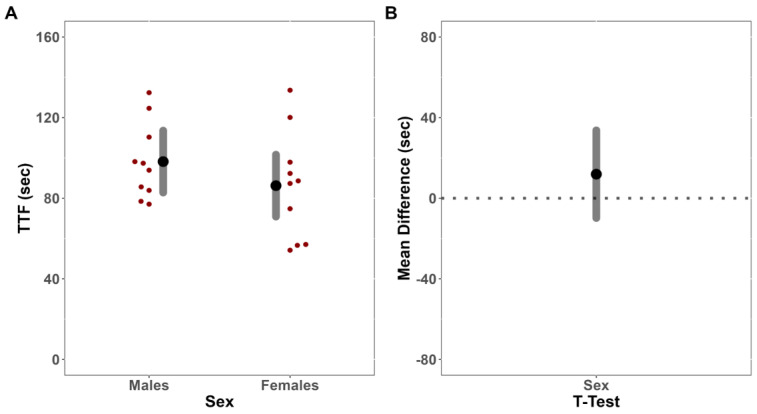
Time-to-task failure (TTF) analysis. (**A**) Estimated marginal means (large black dots) and 95% confidence intervals (CI_95%_) (grey band), paired with individual data (small red dots), for the comparison of TTF between males and females. (**B**) Comparison of mean difference (large black dot) (illustrated with CI_95%_, grey band) for TTE between males and females.

**Figure 2 jfmk-08-00085-f002:**
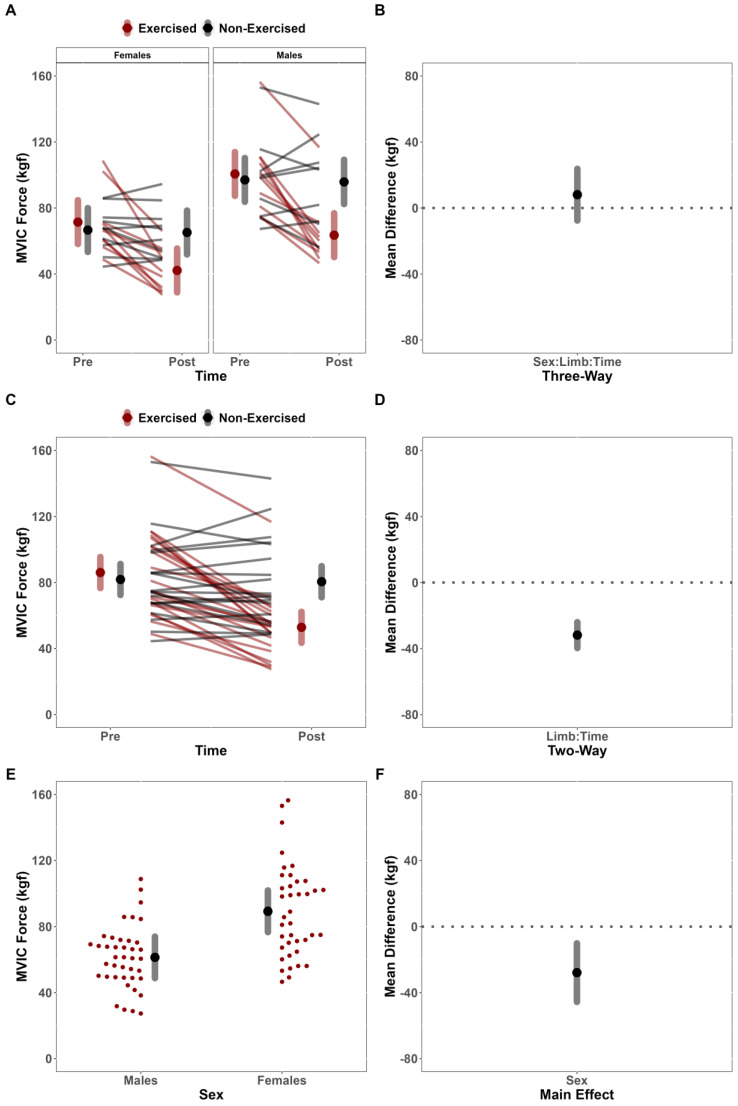
Maximal voluntary isometric contraction force (MVIC) analysis. (**A**) Estimated marginal means (large red/black dots) and 95% confidence intervals (CI_95%_) (red/grey band), paired with individual data (red/black lines), for the changes in MVIC force of the exercised and non-exercised limb, separated by sex. (**B**) Comparison of mean difference (black dot) (illustrated with CI_95%_, grey band) for change in MVIC force between limbs, compared across sex. (**C**) Estimated marginal means (large red/black dots) and CI_95%_ (red/grey band), paired with individual data (red/black lines), for the changes in MVIC force of the exercised and non-exercised limb, adjusted for sex. (**D**) Comparison of mean difference (black dot) (illustrated with CI_95%_, grey band) for change in MVIC force between limbs, adjusted for sex. (**E**) Estimated marginal means (large black dots) and CI_95%_ (grey bands), paired with individual data (small red dots), for the MVIC force of males and females, adjusted for Time and Limb. (**F**) Comparison of mean difference (black dot) (illustrated with CI_95%_, grey band) for MVIC force between Sexes, adjusted for Time and Limb.

**Figure 3 jfmk-08-00085-f003:**
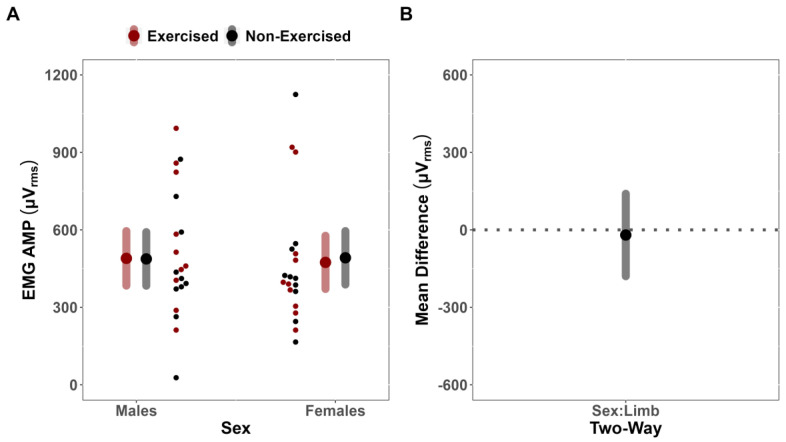
Surface electromyographic amplitude (sEMG AMP) analysis. (**A**) Estimated marginal means (Large red/black dots) and 95% confidence intervals (CI_95%_) (grey/red bands), paired with individual data (small red/black dots), for the effects of fatigue on sEMG AMP of the exercised and non-exercised limb, separated by Sex. (**B**) Comparison of mean difference (black dot) (illustrated with CI_95%_, grey band) for the effects of fatigue on sEMG AMP of the exercised and non-exercised limb, compared across Sex.

**Figure 4 jfmk-08-00085-f004:**
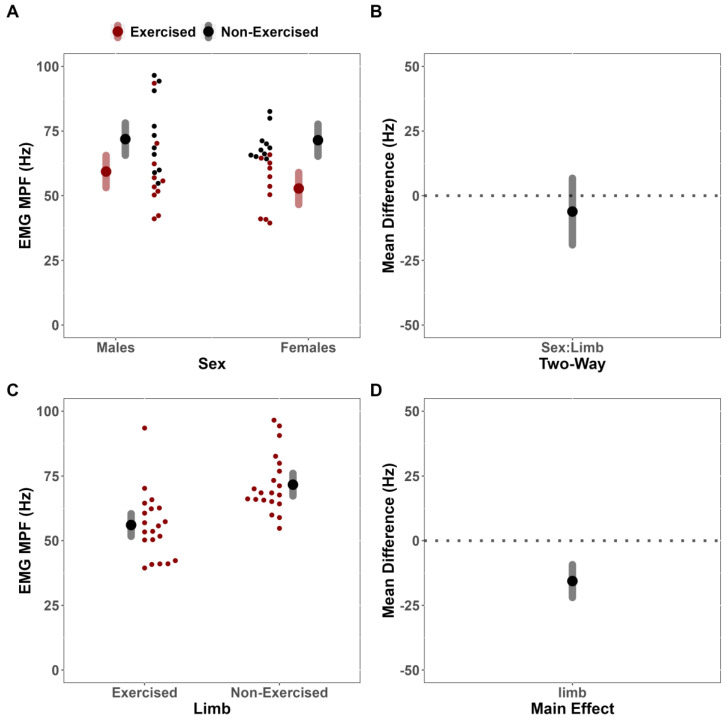
Surface electromyographic mean power frequency (sEMG MPF) analysis. (**A**) Estimated marginal means (large red/black dots) and 95% confidence intervals (CI_95%_) (red/grey bands), paired with individual data (small red/black dots), for the effects of fatigue on sEMG MPF of the exercised and non-exercised limb, separated by Sex. (**B**) Comparison of mean difference (black dot) (illustrated with CI_95%_, grey band) for the effects of fatigue on sEMG MPF of the exercised and non-exercised limb, compared across Sex. (**C**) Estimated marginal means (large black dots) and CI_95%_ (grey bands), paired with individual data (small red dots), for the effects of fatigue on sEMG MPF of the exercised and non-exercised limb, adjusted for sex. (**D**) Comparison of mean difference (black dot) (illustrated with CI_95%_, grey band) for the effects of fatigue on sEMG MPF of the exercised and non-exercised limb, adjusted for Sex.

**Figure 5 jfmk-08-00085-f005:**
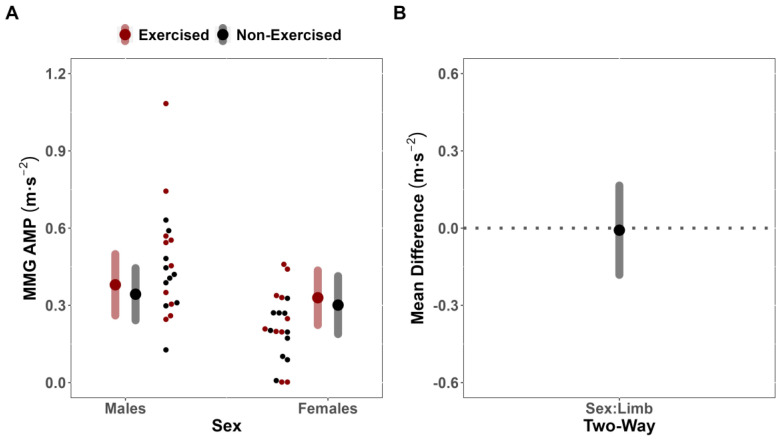
Surface mechanomyographic amplitude (sMMG AMP) analysis. (**A**) Estimated marginal means (large red/black dots) and 95% confidence intervals (CI_95%_) (red/grey bands), paired with individual data (small red/black dots), for the effects of fatigue on sMMG AMP of the exercised and non-exercised limb, separated by Sex. (**B**) Comparison of mean difference (black dot) (illustrated with CI_95%_, grey band) for the effects of fatigue on sMMG AMP of the exercised and non-exercised limb, compared across Sex.

**Figure 6 jfmk-08-00085-f006:**
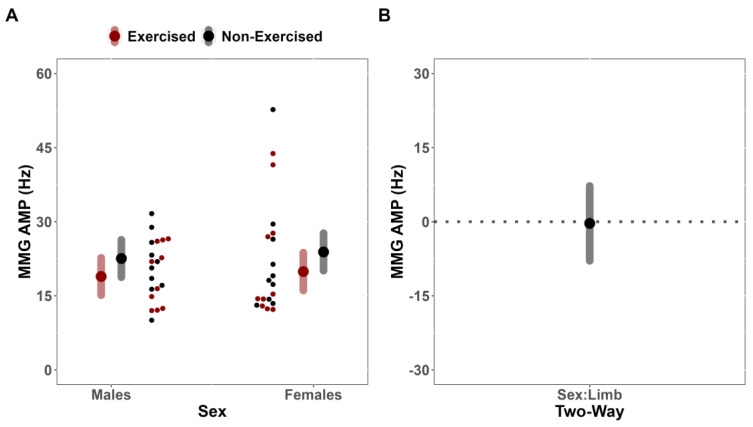
Surface mechanomyographic mean power frequency (sMMG MPF) analysis. (**A**) Estimated marginal means (large red/black dots) and 95% confidence intervals (CI_95%_) (red/grey bands), paired with individual data (small red/black dots), for the effects of fatigue on sMMG MPF of the exercised and non-exercised limb, separated by Sex. (**B**) Comparison of mean difference (black dot) (illustrated with CI_95%_, grey band) for the effects of fatigue on sMMG MPF of the exercised and non-exercised limb, compared across Sex.

**Table 1 jfmk-08-00085-t001:** Participant characteristics presented as mean ± standard deviation.

Measure	Males (*n* = 10)	Females (*n* = 10)
Age (y)	24 ± 3	24 ± 4
Height (cm)	178 ± 8	165 ± 6
Body Mass (kg)	81 ± 15	69 ± 14

Abbreviations: y = years, cm = centimeters, kg = kilograms.

**Table 2 jfmk-08-00085-t002:** Within condition pre- to post-exercise changes for maximal voluntary isometric contraction (MVIC) force and all neurophysiological outcomes.

Sex	Measure	Limb	Pre-Test	Post-Test	%Δ
Males	MVIC Force (KGF)	Exercised	100.64 ± 24.08	63.53 ± 20.09	−36.87
		Non-Exercised	97.02 ± 24.86	93.90 ± 27.21	−3.22
	sEMG AMP (uVrms)	Exercised	641.43 ± 288.13	558.41 ± 256.20	−12.94
		Non-Exercised	501.74 ± 199.45	447.68 ± 237.29	−10.77
	sEMG MPF (Hz)	Exercised	66.69 ± 15.92	57.75 ± 15.22	−13.41
		Non-Exercised	72.71 ± 16.74	73.97 ± 15.25	1.73
	sMMG AMP (m·s^−2^)	Exercised	0.59 ± 0.18	0.51 ± 0.26	−13.56
		Non-Exercised	0.49 ± 0.20	0.41 ± 0.15	−16.33
	sMMG MPF (Hz)	Exercised	24.57 ± 5.08	19.14 ± 6.19	−22.10
		Non-Exercised	23.02 ± 6.02	21.42 ± 6.37	−6.95
Females	MVIC Force (KGF)	Exercised	71.47 ± 19.24	42.16 ± 13.33	−41.01
		Non-Exercised	66.65 ± 13.74	65.17 ± 15.78	−2.22
	sEMG AMP (uVrms)	Exercised	555.51 ± 236.87	476.06 ± 245.57	−14.30
		Non-Exercised	513.29 ± 206.45	460.97 ± 259.56	−10.19
	sEMG MPF (Hz)	Exercised	70.66 ± 10.38	53.63 ± 10.26	−24.10
		Non-Exercised	67.09 ± 4.89	70.11 ± 6.29	4.50
	sMMG AMP (m·s^−2^)	Exercised	0.24 ±0.14	0.24 ± 0.16	0.00
		Non-Exercised	0.20 ± 0.09	0.19 ± 0.10	−5.00
	sMMG MPF (Hz)	Exercised	26.87 ±11.09	22.17 ± 12.22	−17.49
		Non-Exercised	22.79 ±10.90	22.55 ± 11.90	−1.05

Abbreviations: MVIC = maximal voluntary isometric contraction, kgf = kilograms-force, sEMG AMP = surface electromyographic amplitude, uV = microvolts, rms = root mean square, sEMG MPF = surface electromyographic mean power frequency, Hz = hertz, sMMG AMP = surface mechanomyographic amplitude, m·s^−2^ = meters per second squared, sMMG MPF = mechanomyographic mean power frequency, %Δ = mean percent change.

**Table 3 jfmk-08-00085-t003:** Reliability analysis.

Measure	ICC (2,k)	ICC95%	Summary	SEM	CV (%)	MDC
MVIC Force (kgf)	0.9866	0.9751 0.9934	Excellent	7.05	8.77	19.54
sEMG AMP (uVrms)	0.9616	0.9282 0.9811	Excellent	106	19.8	293.82
sEMG MPF (Hz)	0.9298	0.8689 0.9655	Excellent	7.15	10.4	19.82
sMMG AMP (m·s^−2^)	0.9658	0.9367 0.9831	Excellent	0.10	25.4	0.27
sMMG MPF (Hz)	0.9528	0.9123 0.9767	Excellent	4.18	17.7	11.59

Abbreviations—ICC = intraclass correlation coefficient, SEM = standard error of the measurement, CV = coefficient of variation, MDC = minimal detectable change, MVIC = maximal voluntary isometric contraction, kgf = kilograms-force, sEMG AMP = surface electromyographic amplitude, uV = microvolts, rms = root mean square, sEMG MPF = surface electromyographic mean power frequency, Hz = hertz, sMMG AMP = surface mechanomyographic amplitude, m·s^−2^ = meters per second squared, sMMG MPF = surface mechanomyographic mean power frequency.

## Data Availability

The model outputs and corresponding codes used for this analysis are available on the Open Science Framework project page for this study (https://osf.io/cmzqx, accessed on 2 May 2023).

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
