# Peer review of "Unilaterally Induced Quadriceps Fatigue during Sustained Submaximal Isometric Exercise Does Not Alter Contralateral Leg Extensor Performance"

_jfmk, 2023, doi:10.3390/jfmk8020085_

Round 1

Reviewer 1 Report

Is it the case that your finding of no effects of isometric exercise is due to the intensity (i.e. 50%) of the MVC. It is likely that the primary cause for fatigue was peripheral with limited, if at all, central contributions. I suggest to revise the title and be more specific with the “isometric exercise” phrase.

In addition, the first paragraph of the introduction could clarify under what circumstances (i.e. what kind of exercise with duration and intensities) is there a meaningful contribution of the central factors to the exercise-induced fatigue.

Please provide the justification for using a 50% MVC and the muscle group of interest. Does the literature show that females are more fatigue-resistance with that muscle group and intensity?

L8. Please revise as it seems observations are on the performance in both legs following unilateral fatiguing isometric exercise and MVCs were done in the exercise and non-exercised legs on task failure. See also Ls 101-103.

L15. Is it correct that the p-values are the same.

L19. The statement “There were no differences in time-to-task failure or performance fatigability between males and females” is a repeated of what is stated as well in Ls 14-15.

L20. Similar for the statement “Moreover, the non-exercised limb's performance 20 was not affected following the unilateral fatigue intervention.” That feels like a repeat.

Ls 21-23. It is not clear whether this is a conclusion on a sex-effect. Please clarify.

L50. What was the intensity of the exercise modality in Neltner et al.

L53. What was the intensity of the exercise in Martin and Rattey.

L63. What was the intensity of the exercise in Doix et al.

L66. What was the intensity in Voskuil et al.

In L70, is mentioned the potential role of the intensity so please provide more detailed information on the studies covered in the introduction that examined contralateral fatigue, as suggested above as well.

L77. You state “greater centrally-mediated contralateral fatigue for males compared to females”, but later in the sentence there is mention of no change in females. Please clarify.

L140. Do we need in the legend description of the abbreviations?

L167. If the post-testing was initiated immediately after task failure with the 50%MVC and 3 maximal 3 seconds contractions within 15 seconds, then how can the authors be sure that that was not adding the fatigue already experienced especially when the first of the 3 contractions would be the bilateral isometric contraction. I suggest to provide a comment in the discussion.

L168. It seems the data on the bilateral contraction after task failure are not presented. Why was this contraction done? Please clarify.

I suggest to move Table 2 to the results section.

In the legend of Table 2, define rms.

Figure 2E is unclear. Why plotting only red dots (assuming exercised leg) and mean and CI for non-dominant. Please clarify. See also Figure 4C.

Figure 2 and text. Would it not be better to present the data as exercised-dominant and non-exercised non-dominant.

Ls 389-390. At what intensity of the isometric contraction is there occlusion and basically no oxygen delivery, see DOI: 10.1249/mss.0b013e31802dd3cc.

L454. Is it possible at all that a non-exercised leg experiences peripheral fatigue? This seems an odd statement.

L509. Was the bilateral always performed before the non-dominant as it seems the 3 post failure contractions (i.e. bilateral, dominant and non-dominant) were in random order according to L169. Please clarify.

Please ensure a consistent referencing format according to journal guidelines.

Reviewer 2 Report

The reviewed manuscript discusses the impact of quadriceps femoris fatigue on the contralateral limb extensors. The main purpose of this study was to investigate the effect of submaximal, fatiguing, isometric leg extensions on MVIC force of the contralateral, unexercised limb in male and female. Additionally, the Authors tried to find the potential neurophysiological mechanisms related to a cross-over effect.

The introduction is very extensive. Issues related to the topic and the research problem are described in an exhaustive way. However, part of the introduction seems to be better suited to the discussion chapter. First of all, those fragments in which the authors cite in detail the results obtained by other researchers. I think that in the introduction it is enough to signal them without detailed analysis.

The study group and the methods of assessing muscle fatigue and interpreting the results are very thorough and extensive. A detailed description of the research procedure will make it easier to relate the results of the experiment to other research. The choice of methods for a specific purpose of work is correct.

In the chapter devoted to research results, there is a very large amount of information, which makes it difficult to receive (read?) and understand this part of the manuscript. The charts contained in the work illustrate corectly the information contained in the text. I miss the drawing with a fragment of the EMG recording showing the tested parameters. Table 2 contains too much data. Maybe it can be divided into two smaller tables, e.g. by gender, or graphically presented in a different way.

Discussion is a strong point of the work. The authors compare the results of their own research with numerous previous reports. The literature is rich, up-to-date and correctly cited.

I believe that the work is valuable and can be published after correction.

Round 2

Reviewer 1 Report

Thanks for addressing all my comments and suggestions.